# Deep Learning-Based Road Traffic Noise Annoyance Assessment

**DOI:** 10.3390/ijerph20065199

**Published:** 2023-03-15

**Authors:** Jie Wang, Xuejian Wang, Minmin Yuan, Wenlin Hu, Xuhong Hu, Kexin Lu

**Affiliations:** 1School of Electronics and Communication Engineering, Guangzhou University, Guangzhou 510006, China; 2Research Institute of Highway Ministry of Transport, Beijing 100088, China; 3National Environmental Protection Engineering and Technology Center for Road Traffic Noise Control, Beijing 100088, China; 4National Engineering Laboratory for Digital Construction and Evaluation of Urban Rail Transit, China Railway Design Corporation, Tianjin 300162, China

**Keywords:** traffic noise annoyance, deep learning, transfer learning

## Abstract

With the development of urban road traffic, road noise pollution is becoming a public concern. Controlling and reducing the harm caused by traffic noise pollution have been the hot spots of traffic noise management research. The subjective annoyance level of traffic noise has become one of the most important measurements for evaluating road traffic pollution. There are subjective experimental methods and objective prediction methods to assess the annoyance level of traffic noise: the subjective experimental method usually uses social surveys or listening experiments in laboratories to directly assess the subjective annoyance level, which is highly reliable, but often requires a lot of time and effort. The objective method extracts acoustic features and predicts the annoyance level through model mapping. Combining the above two methods, this paper proposes a deep learning model-based objective annoyance evaluation method, which directly constructs the mapping between the noise and annoyance level based on the listening experimental results and realizes the rapid evaluation of the noise annoyance level. The experimental results show that this method has reduced the mean absolute error by 30% more than the regression algorithm and neural network, while its performance is insufficient in the annoyance interval where samples are lacking. To solve this problem, the algorithm adopts transfer learning to further improve the robustness with a 30% mean absolute error reduction and a 5% improvement in the correlation coefficient between the true results and predicted results. Although the model trained on college students’ data has some limitations, it is still a useful attempt to apply deep learning to noise assessment.

## 1. Introduction

With the development of urban road traffic, traffic noise pollution has become an increasing public concern [1], which may bring different negative effects on people’s mental and physical health [2,3,4], such as hearing impairment, emotional irritability, and heart disease. Controlling and reducing the damage caused by traffic noise pollution has been the main focus of traffic noise management research [5,6,7], and the annoyance level caused by traffic noise has become one of the important measurements for managing road traffic pollution [8].

The current methods of noise annoyance assessment are mainly subjective and objective: subjective tests usually require social surveys or listening tests [9,10], and social surveys are used to obtain the results of subjects’ evaluation of environmental noise annoyance by means of questionnaires or interviews and are generally used for long-term annoyance assessment. Meanwhile, listening tests are generally used for short-term annoyance assessment [10], which requires replaying noise in a laboratory environment and obtaining the feedback from subjects on their own perceived annoyance. The objective method usually uses the sound pressure level as a measurement indicator [11]. In more advanced annoyance evaluation methods, acoustic features such as loudness, roughness, and sharpness are also applied [12,13,14], which are easy to implement without large-scale subject participation but are usually less reliable than the subjective experimental approach.

Due to the development of artificial intelligence technology, many scholars have also tried to use neural network algorithms for a noise annoyance assessment. Jesus [15] used deep convolutional neural networks to calculate the psychoacoustic annoyance of urban noise, using 1 s noise segments as the input and the psychoacoustic annoyance values of noise as labels, and the model could give appropriate psychoacoustic annoyance assessment results after a long iteration. Song [16] relied on subjective listening experiments to obtain subjects’ perceived noise annoyance datasets and used recurrent neural networks to model perceived noise annoyance, which required complicated features such as MFCC (Mel Frequency Cepstral Coefficient, a kind of speech feature parameter) and noise loudness. Luis [17] used fully connected neural networks to model the social survey results to assess long-term noise annoyance.

This paper follows the idea of objective assessment and proposes a model for the evaluation of road traffic noise annoyance with the deep learning model. The model is based on the results of subjective listening experiments and directly constructs the mapping relationship between noise fragments and annoyance, without extracting additional acoustic features, which can achieve a fast evaluation of noise annoyance. However, due to the long data acquisition period of subjective listening experiments, the available sample size is small at this stage, and problems such as over-fitting easily occur in the training process [18]. To avoid this problem, this paper introduces transfer learning to solve this problem. Transfer learning can be divided into homogeneous and heterogeneous types [19,20], and homogeneous transfer learning is often used in tasks with similar data domains, while heterogeneous transfer learning is often used in tasks with different data domains. Since there is a great correlation between the psychoacoustic annoyance level of noise and the perceived noise annoyance level obtained from listening experiments, the connection between them can generally be established by logistic regression [21], and the samples required for calculating the psychoacoustic annoyance level and listening experiments are from the same data domain, so the feature transfer of the psychoacoustic annoyance level dataset can be applied to the evaluation of the perceived noise annoyance level. Following the transfer learning strategy [19,20], the method does not require additional input information and keeps the computational complexity of the algorithm unchanged. Second, compared to the way of increasing the amount of data to obtain features, there is an objective method for calculating the psychoacoustic annoyance of noise [21], and it is only necessary to produce the dataset with the help of a computer, by transferring the psychoacoustic feature into the deep learning model, which does not require large-scale listening experiments.

Based on the experimental results of this paper, the method using deep learning has reduced the mean absolute error of the prediction by about 30% more than traditional machine learning algorithms such as the regression algorithms and neural network; however, the performance is still poor in the annoyance interval with few samples. The transfer learning strategy can effectively moderate the over-fitting phenomenon, such as bad performance in few-samples intervals, but performs well in large-samples intervals [22], which greatly improves the robustness of the algorithm. The structure of this paper is as follows: the first part introduces the listening experiment-related content and dataset construction; the second part introduces the deep learning model architecture and experimental setup; the third part gives the results and analysis; the fourth part is the discussion; the final part is the conclusion.

## 2. Subjective Listening Experiment and Dataset Construction

### 2.1. Listening Experiment Dataset

The audio data for the listening experiment came from two parts: one from the SoundIdealGeneral6000 sound library [23] resampled to 48 kHz. Another part came from the road collection, as shown in Figure 1, and the collection location was the Huangpu Avenue in Guangzhou City, Guangdong Province. The data were collected at several places on Huangpu Avenue in the morning and evening. The road surface was asphalt, the continuous equivalent sound pressure level of the collected noise was concentrated in the range of 60–65 dBA, the road traffic flow was concentrated in the range of 400 to 800 vehicles/h, the wind speed was less than 1 m/s, and the temperature was 17–25 °C. In order to reduce the impact of reflected sound on recording, we had to ensure that the recording equipment did not contain large reflectors (such as walls) within 10 m. If the ambient sound contained too many additional sound sources, such as the sound of birds and the sound of pedestrians, the recording needed to be stopped in time and the recording results should not be used in the listening experiment. The acquisition equipment song meter4 (SM4) [24] was used for road noise recording, which was about 4 m away from the center line of the outermost lane, and the equipment was about 1.5 m high. The recorded data were two-channel, and the sampling frequency was 48 kHz. The total recording duration was 30 h.

The recorded data were separated into 8 s clips. The average loudness of the audio data was between 55 phon and 90 phon. Due to the lack of a motorcycle sound, whistle road sound, road sound on rainy days, and so on, we selected the appropriate data from the sound library. A total of 949 audio data were collected.

### 2.2. Listening Experiment Settings

The experiments were conducted in an audiometric room ($6.78\times 3.51\times 2.26\;\left(m\right)$) with a background noise of less than 25 dBA, and the walls and floors of the audiometric room were covered with sound-absorbing materials. The noise was passed through a high-quality sound card (RME Fireface II) [25] for playback and then subjects perceived the noise audio through headphones HD600 [26]. The audio playback process was written in Python and the subjects operated the interface themselves for audio playback and scoring. During the listening process, the audio was played at random, and each audio would be played 3 times. A number of noises were played at random before the experiment started so that the participant could familiarize themselves with the interface and the process. To ensure the validity of the results, each listening session was strictly limited to 40 min and a total of 14 rounds were conducted. To ensure an accurate assessment of noise, subjects were allowed to stop and rest for 2–3 min during the listening experiments. The same subject conducted the listening experiment at the same time of the day.

The annoyance scale was based on the 11-level assessment scale of the ISO 15666:2021 standard [27], in which subjects were asked to choose a value from 0 to 10 to characterize their annoyance level after being exposed to noise stimuli, where 0 represented no annoyance and 10 represented extreme annoyance.

A total of 20 subjects, 15 males and 5 females, aged between 20 and 32 years, were invited to participate in the listening experiment. All subjects had normal hearing. The subjects were paid for completing the experiment. The noise feature data could be used as the algorithm input and the subjects’ post-processed annoyance values were used as the label; this built a supervised learning dataset suitable for training algorithms. Although the dataset was literally enough for the algorithm, there were still some issues that need to be mentioned: (1) with the uneven distribution of subjects, the elder should be considered. Due to the lack of annoyance data about old people, this will lead to changes in the subjective annoyance value. (2) The number of subjects: referring to documents [28,29,30], 20 subjects can basically provide feedback on the annoyance of noise data, but the more subjects, the better [17,31,32,33]. (3) Limitation of laboratory playback: laboratory playback can recreate the noise well, but it definitely did not recreate the whole of the experience. The laboratory environment did not simulate the subject’s situation at the roadside. For a wider review and more complete subjective assessment, a qualitative interview is essential [10,17,34,35]. (4) The recording scene was not rich enough. In this paper, the authors just recorded the noise data on the same avenue; therefore, noise from other sites should also be taken into consideration.

### 2.3. Listening Experiment Results

In the listening experiment, each noise sample was played three times at random. If the difference in annoyance given by the same subject for any two of the three evaluations of the same noise sample was 2 or more, the evaluation sample would be regarded as misjudged and needed to be removed. If the number of one subject’s misjudged samples reached 30% or more of the total sample size, the experimental results of that subject needed to be removed.

After removing the invalid data, the results of the three scoring sessions of the noisy samples were averaged as the subject’s perceived annoyance with the noise samples. If the number of perceived annoyance assessments for a noise sample was less than 14, the noise sample needed to be excluded.

After eliminating the invalid samples, the mean annoyance (MA) of the noise samples was calculated as the model label. In this paper, we selected the amplitude spectrum as the model input. The amplitude spectrum can directly provide feedback on the characteristic information of the noise; it included the amplitude value of the noise at different frequencies. To obtain the amplitude spectrum, we should conduct an STFT (short time Fourier transform, a method to determine the frequency and phase of a sine wave in the local area of a time-varying signal) on noise first, and then calculate the modulus of STFT results to obtain the amplitude spectrum. The process of STFT as a nonlinear transformation and formula is as follows:(1)$$\mathrm{STFT}\left(t,f\right)=\int_{-\infty }^{+\infty }\left[x\left(\tau \right)w\left(\tau -t\right)e^{-j2\pi f\tau }\right]d\tau$$
where $w\left[\;\right]$ is the window function,$\;x\left[\;\right]$ is the noise signal, t is the time index, τ is the interval with time t, and f is the frequency index. The output of STFT is a complex number, and then the amplitude spectrum is calculated by the STFT results of the noise signal, where the STFT window is the hamming window, the window length is 512, and the jump length is 256. The calculation of the amplitude spectrum is shown as follows:(2)$$\mathrm{Amplitude}\;=\sqrt[{2}]{R{\left(\mathrm{STFT}\left(t,f\right)\right)}^{2}+I{\left(\mathrm{STFT}\left(t,f\right)\right)}^{2}}$$
where R is the real part of $\mathrm{STFT}\left(t,f\right)$ and I means the imaginary part of $\mathrm{STFT}\left(t,f\right)$. The function of the hamming window is as follows, where *N* is the window length minus 1:(3)$$w\left[n\right]=0.54-0.46\cos(2\pi \frac{n}{N}),\;0\;ࣘ\;n\;ࣘ\;N$$

Due to calculating the mean value of annoyance, the annoyance value is no longer an integer type, so this paper divided the annoyance range [0–10] into different annoyance intervals with an interval size of 1.949 samples obtained, and the distribution of the samples is shown in Table 1. The level of annoyance caused by traffic noise is usually high; therefore, few samples fall in the [0,3) interval and the most noise samples fall in [3,9).

### 2.4. Extended Dataset

Deep learning is a data-driven approach; therefore, training on a small dataset can lead to over-fitting. To avoid the bad effects of inadequate listening experimental samples on the deep learning model, this paper used transfer learning to extract the features of numerous samples. Due to the correlation between the psychoacoustic annoyance level of noise and the perceived noise annoyance [22], we built a psychoacoustic annoyance dataset to realize the feature transferring. The extended data sources were the same as in Section 2.1. After excluding the raw data used in the listening experiments, the remaining noise data were cropped into 8 s clips and sampled at 48 kHz. The cropped data were normalized to the full scale to produce noise data; the mean psychoacoustic annoyance was calculated by the loudness, sharpness, roughness, and fluctuation of the two channels as labels [22] to construct the psychoacoustic annoyance dataset; and a total of 17,025 data were obtained. An STFT was performed on the noise, and the amplitude spectrum was calculated as the model input, where the window was the hamming window, the STFT window length was 512, and the jump length was 256. The distribution of the data samples is shown in Table 2. The samples should be evenly distributed in each interval as far as possible, where the psychoacoustic annoyance interval meant the psychoacoustic annoyance value in different intervals, and the numbers meant the amount of samples in the corresponding interval. The psychoacoustic value was the continuous value, and the most data fell in (0,90].

## 3. Research Method

Common approaches to transfer learning in deep learning are overall parameter optimization based on pre-trained models [36,37] and partial parameter fine-tuning based on pre-trained models [38,39,40]. Overall parameter optimization based on pre-trained models refers to the initial training of the model parameters in an additional data domain to obtain a pre-trained model, followed by a secondary training of the overall parameters of the model in the data domain required for this task. Fine-tuning, on the other hand, involves the secondary training of only some of the parameters within the data domain required for this task, with the other parameters fixed. In this paper, we first use the psychoacoustic annoyance dataset for pre-training. The amplitude spectrum of the noise is used as the input and the psychoacoustic annoyance level of the noise is used as a label for the initial optimization of the model parameters. The final road traffic noise annoyance assessment model is obtained by using the listening experiment dataset to optimize the overall parameters and some parameters of the pre-trained model, respectively.

### 3.1. Model Architecture

With the development of deep learning, a large number of researchers have tried to use deep learning on their own tasks, and many researchers have successfully applied deep learning algorithms in many of the fields such as image segmentation [41], speech enhancement [42], hearing aids [43], traffic prediction [44], and so on. In this process, many classic deep learning model architectures have been gradually created, among which the most widely used and effective model is UNet [45]: almost all of the tasks achieved good results by using UNet architecture or modifying the UNet architecture according to the requirements of the tasks [41,44,45]. The model extracts deep features of the data by adding a convolutional downsampling module [41] to the basic convolutional neural network [46] and recovers the dimensionality of the data through a convolutional upsampling [41] module.

In this paper, we use UNet as the basic model and the reasons are as follows: (1) UNet and its variant’s version have been used in many research areas with good results, and it can provide ideas for our research [41,44,45,47]. (2) UNet is a kind of deep convolutional neural network [46], and the convolutional operation [46] is commonly used in the deep learning model. By using the convolutional operation, maxpooling operation, and activate function, the network can implement nonlinear transformations to obtain the feature map. (3) UNet is a mature deep learning network architecture. Compared with other model structures, the structure of UNet is relatively simple, it has more online open source information, and it is easy to reproduce [48]. In this paper, by modifying the structure according to the task requirement, we enable an end-to-end evaluation of traffic noise based on the deep learning model. The model does not need to recover the output dimension of the data but only needs to use the convolutional downsampling module for the feature extraction of road noise to obtain the feature map. The feature fusion module will fuse the extracted features and then use the fully connected layer network [46] as a decoder to output the final result. The model architecture is shown in Figure 2.

#### 3.1.1. Encoder

The encoder uses a two-dimensional convolution operation with a convolution kernel of 1×1 to expand the input information to eight dimensions to establish proper feature mapping paths in the high-dimensional space of feature learning. The input is noted as x and the shape is $\left(B,1,F,E\right)$, where B is the number of samples in the same batch, F is the length of each input data, and E is the width of each input data. The output of the encoder is as follows:(4)$$x_{\mathrm{encode}}=x*W+b$$
where W denotes the convolution filter weights and b is the bias, the ∗ represents a convolution and the convolution operation is denoted as Conv2d(.) in the next description. The shape of the output is $\left(B,8,F,E\right)$.

#### 3.1.2. Convolutional Downsampling Module

The convolutional downsampling module realizes the downsampling function of information through a two-dimensional convolution operation and maximum pooling operation to facilitate the acquisition of detailed information of data. The convolutional downsampling will output the feature extraction result and downsampling result, the operation process of the left and right channel is the same, but the convolutional parameters are not shared: by taking one of the ways as an example, you can note the output of i−th convolutional downsampling module is ${\mathrm{Out}}_{i}$. The input to the convolutional downsampling module is denoted as $X_{\mathrm{in}}$, and the shape is $\left(B,C_{\mathrm{in}},F,E\right)$. The output of the convolutional downsampling module is as follows:(5)$$\mathrm{Convout}1=\sigma (\mathrm{Conv}2d(X_{\mathrm{in}}))$$
where the $\sigma \left(.\right)$ denotes the LeakyReLU operation to scale the value range into $\left(-\infty ,1\right)$.
(6)$$\mathrm{Convout}2=\sigma \left(\mathrm{Conv}2d\left(\mathrm{Convout}1\right)\right)$$
(7)$${\mathrm{Out}}_{i}=\mathrm{MaxPool}2d\left(\mathrm{Convout}2\right)$$

The shape of ${\mathrm{Out}}_{i}$ is $\left(B,C_{\mathrm{out}},F/2,E/2\right),$ and the step of maximum pooling is (2,2).

#### 3.1.3. Feature Fusion Module

The feature information obtained through convolutional downsampling interacts with other features through the concatenating operation, and the fusion of the two-feature information is realized through the feature fusion module to provide input for the decoder. The input of the left path is noted as $L_{\mathrm{out}}$ and the input of the right path is noted as $R_{\mathrm{out}}$ where ⊕ denotes the concatenating operation in the channel dimension. The output is as follows:(8)$$\mathrm{TempOut}=\mathrm{Conv}2d\left(\mathrm{Lout}⊕\mathrm{Rout}\right)$$
(9)$$\mathrm{MixOut}=\mathrm{Conv}2d\left(\mathrm{Conv}2d\left(\mathrm{Conv}2d\left(\mathrm{Tempout}\right)\right)\right)$$

#### 3.1.4. Decoder

The output after the feature fusion module is sent to the decoder to calculate the final result. To better correspond the model output to the annoyance value, unlike the previous use of fully connected operations to map features from high to low dimensions, this paper performs separate dimensional reduction operations in different dimensions to take into account the different feature dimensions. The final output of the model is as follows:(10)$$\mathrm{FreqOut}=\sigma \left(\mathrm{Linear}1\left(\mathrm{Out}\right)\right)$$
(11)$$\mathrm{FinalOut}=\mathrm{Linear}2\left(\mathrm{FreqOut}\right)$$

### 3.2. Model Parameter Setting and Optimization

As shown in Figure 2, the number of channels of the model is set as follows: C1 = 32, C2 = 64, C3 = 32, and C4 = 8. The numbers of output channels of the feature fusion module are 16, 8, and 1, respectively. The hidden nodes of the linear layer of the decoder are 16 and 93, respectively, and the detailed settings of the model are shown in Table 3. The batch size is set to 5 and the epoch is set to 100. The pre-train stage and the secondary optimization stage both use the model output, and the mean square error of the true labels is used as the loss function, and the optimizer is Adam [49] with an initial learning rate of $1\times 10^{-3}$. A cosine update strategy is chosen to dynamically adjust the learning rate, and the cut-off size of the learning rate is $1\times 10^{-7}$.

If the model does not decrease in loss value after 10 iterations of updates, the training is withdrawn early. The model parameter with the lowest loss in the validation set is saved as the final training result.

## 4. Experimental Results and Analysis

### 4.1. Pre-Training Stage

#### 4.1.1. Pre-Training Dataset Setup

Generally, for machine learning and deep learning datasets, the number of training sets is 70–80% of the total samples, and the number of samples in the validation and test sets is 20–30% of the total samples [50,51]. Take the MNIST [52], for example, which is a common dataset in machine learning as well as deep learning: this dataset has a training set of 55,000 (78% of total samples), a validation set of 5000 (8% of total samples), and a test set of 10,000 (14% of total samples). If the samples in the training set are not sufficient, it will lead to an over-fitting phenomenon [53], and if the samples in the training set are overdose, it is difficult to measure the ability of the model to handle unknown samples in the future. At the pre-train stage, the model parameters will be initially optimized on the psychoacoustic dataset (Section 2.4, Table 2), the total samples are 17,025, 12,910 (75% of the total samples) data are used for training, 1600 (10% of the total samples) data for validation, and 2515 (15% of the total samples) data for testing. The model which has the best performance on the validation set will be saved as the pre-trained model.

#### 4.1.2. Pre-Training Model Results

The psychoacoustic annoyance ranges from 0 to 100, and the model output will multiply a number to suit the range. In this paper, the best number is 80 according to the training experience. The complexity of the pre-training model is measured by the number of floating point operations, and it is 5.7 GMac. The number of hyperparameters is 0.203 M. The model with the lowest loss in the validation set is selected as the pre-training model. In this paper, we select mean absolute error (MAE) as an evaluation standard. The MAE can visually provide feedback on the average error between the predicted result and the real result, the evaluation weight for each error is equal, it is less affected by anomaly samples [54], and it suits evaluating the overall performance of the model. The formula of MAE is as follows:(12)$$\mathrm{MAE}=\frac{1}{N}\overset{N}{\underset{i=1}{\sum }}\left|x_{i}-y_{i}\right|$$
where N is the number of all samples,$\;x_{i}$ is the predicted value of the i−th sample, and $y_{i}$ is the label of i−th sample. The MAE of the pre-training model ranges from 0 to 100. The error in the training set, validation set, and test set are 3.35, 4.03, and 3.25, respectively.

### 4.2. Formal Training Stage

#### 4.2.1. Formal Training Dataset Setup

At the formal training stage, based on the pre-trained model, the model parameters will be optimized on the listening experiment dataset (Section 2.3, Table 1). The 949 data from the listening experiment are divided into 669 (70% of the total samples), 70 (7% of the total samples), and 210 (23% of the total samples) for training, validation, and testing, respectively. To avoid analysis errors due to the small amount of test data, this paper chooses the best model in the validation and analysis of the performance on the mixed set (mix test data with validation data). The distribution of the mixed dataset is shown in Table 4. In this mixed set, no value falls in [0,2) and [9,10).

#### 4.2.2. Formal Training Results and Comparison

This section performs a secondary optimization of the overall parameters and the model’s decoder parameters, respectively. The two methods will be denoted as total-tuning and fine-tuning. The training dataset is the listening experiment dataset with the total. Due to the different value ranges, in the secondary optimization, the model output will multiply eight by the empirical to suit the range. To compare the performance of the algorithms, artificial neural networks [17], linear regression [17], Lasso regression, and Ridge regression [16], and directly trained models, denoted as direct, are used as comparison algorithms. Different from the algorithms based on deep learning, the machine learning algorithm includes an artificial neural network, and regression algorithms should conduct feature engineering first. In this paper, the input information is the amplitude spectrum, and we use a principal component analysis (PCA) to achieve feature dimension reduction and then input the post-processed feature vectors. The PCA obtains the first 28 principal eigenvalues, and a total of 56 features (left channel and right channel) are obtained. The artificial neural network has nonlinear fitting capabilities; in this paper, the built artificial neural network consists of five linear fully connected layers and the output of each layer is activated by LeakyReLU. The regression algorithms are common analysis methods [16,17,50]. It is difficult to directly figure out the relationship of post-processing features, so this paper uses linear regression, Lasso regression, and Ridge regression as the basic comparison algorithms to evaluate the noise annoyance. The MAE is chosen to measure the error between the evaluation results and true results. Pearson’s correlation coefficient (Formula (13), denoted as PCC) [54], and Spearman’s correlation coefficient (Formula (14), denoted as SCC) [54] are chosen to measure the correlation between the evaluation results of the model and the true results.
(13)$$\mathrm{PCC}=\frac{\mathrm{cov}(x,y)}{\sigma_{x}\sigma_{y}}$$
where x, y are two different arrays, cov(.,.) is the covariance matrix, $\sigma_{x}$ is the standard deviation of x, and $\sigma_{y}$ is the standard deviation of y.
(14)$$\mathrm{SCC}=\frac{\sum_{i=1}^{N}\left(x_{i}-\;\bar{x}\right)\left(y_{i}-\;\bar{y}\right)}{\sqrt{\sum_{i=1}^{N}{\left(x_{i}-\;\bar{x}\right)}^{2}.\sum_{i=1}^{N}{\left(y_{i}-\;\bar{y}\right)}^{2}}}$$
where x, y are two different arrays, N is the total number of each array, $\bar{x}$ is the mean value of x,$\;\mathrm{and}\;\bar{y}$ is the mean value of y.

Figure 3 shows the MAE changes of deep learning-based methods during algorithm training. The initial error of the direct method is greater than fine-tuning and total-tuning, and this may be affected by transfer learning. Fine-tuning and total-tuning perceive the characteristics of psychoacoustic annoyance in advance, and the error of fine-tuning and total-tuning in each iteration is always less than direct.

The evaluation accuracy of each type of algorithm on different annoyance intervals is shown in Table 5 and Table 6 and demonstrates a comparison of the overall evaluation performance of the three strategies.

A comparison of the evaluated performance of each algorithm on different annoyance intervals is shown in Table 5. In terms of the entire mixed dataset, compared to the mean absolute error of regression algorithms and the neural network, the mean error of direct is just 0.57, and the direct reduces the mean error by about 30%. Generally, the deep learning model performs best overall. In the annoyance intervals [2,5) and [6,9), total-tuning, fine-tuning, and direct obtain better results than all types of regression algorithms and artificial neural network algorithms. While in the interval [5,6), regression algorithms have the smallest evaluation error, which is better than direct, total-tuning, and fine-tuning, and within the interval [6,7), artificial neural networks have the most accurate assessment with an error of 0.26. The main reason for this phenomenon is the uneven distribution of the number of samples in different intervals. Both regression algorithms and artificial neural network algorithms prioritize learning the features present in the majority of samples. However, taking into account the information regarding the features of a small number of samples becomes challenging, leading to a significant error when evaluating small sample intervals. Although the accuracy of evaluation in larger sample intervals is high, the overall performance remains unsatisfactory, resulting in a severe over-fitting issue. Direct, total-tuning, and fine-tuning are algorithms that use deep learning models with strong feature-learning capabilities and perform better than regression and artificial neural network algorithms, but there are differences in the results due to the different training methods. Although direct methods have a significant improvement in the interval [2,3) compared to regression algorithms and artificial neural networks, total-tuning and fine-tuning use transfer learning to further reduce the evaluation error and outperform the direct method in all annoyance intervals. In terms of the difference between the maximum and minimum assessment errors for the different annoyance intervals, direct is 1.21 (maximum is interval [2,3), minimum is interval [4,5)), while fine-tuning and total-tuning are 0.45 and 0.38, respectively, and the algorithm using transfer learning greatly improves the robustness of the assessment.

Table 6 shows the MAE, PCC, and SCC results in the mixed dataset (data shown in Table 3). Total-tuning has the smallest evaluation error, but it is close to that of fine-tuning. On the other hand, compared to the MAE of direct, the fine-tuning is 0.45. Fine-tuning reduces this error by about 30%, and the evaluation results obtained by optimizing with the fine-tuning strategy have the largest correlation with the true results. The Pearson correlation coefficient and the Spearman correlation coefficient reach 0.93 and 0.92, compared to the Direct, they improve by about 6% and 5% respectively, which show that the algorithm trained by transfer learning has a larger overall improvement than direct training.

## 5. Discussion

In this paper, a deep learning-based objective assessment method for road traffic noise annoyance is proposed. Different from [16] and [17], this method does not need subjects’ personal information or additional acoustic features. By inputting the amplitude spectrum, this assessment method can rapidly evaluate the annoyance level of road noise. To train the model, twenty subjects (fifteen males and five females, aged between 20 and 32 years) were paid for joining in a listening experiment. From the perspective of algorithm training, the amplitude spectrums of noises are input data, and subjects’ annoyance scores are used as the label. This builds a supervised learning dataset suitable for training algorithms, and the results of the different algorithms show the feasibility of training on this dataset. However, it should be noted that the 20 subjects can basically provide feedback on the annoyance of noise data [28,29,30], but the more subjects join in the listening experiment, the more the results would be closer to the real world [31,32,33]. The distribution of subjects is equally important: lacking the old’s subjective results, this approach may fail in the elderly population. The college students are mainly young people, and they often have normal hearing ability. They can perceive traffic noise in the full frequency band; however, as people get older, they often suffer from hearing loss [55,56], and their ability to perceive a certain frequency of noise will decline. In other words, these people cannot perceive traffic noise in the full frequency band. This direct difference in perception can lead to a bias in final judgment. Some researchers [57] investigated the effect of age on speech perception in noisy environments. The researchers compared the performance of college students to that of older adults. They found that college students had better speech perception in noise than older adults. The study [58] also found that college students had greater sensitivity to high-frequency sounds compared to older adults, which may be related to age-related hearing loss. Meanwhile, considering the hearing differences between men and women [59], men tend to lose their ability to hear at a higher-frequency level and women mostly lose their hearing in the lower-level frequencies. The noise with energy concentrated on high frequency may make women feel more upset and the low-frequency noise often make men feel uncomfortable. In addition, it is difficult to reproduce the complete experience by recreating noise in a laboratory environment; therefore, for a more accurate evaluation, a qualitative interview is needed.

This paper adopts three strategies to train the deep learning network, one of which is directly trained on the listening experiment dataset with better performance than commonly used machine learning algorithms. The other two, called total-tuning and fine-tuning, respectively, are pre-trained on the psychoacoustic dataset first and then trained on the listening experiment dataset, where total-tuning will optimize all parameters and fine-tuning just optimizes the parameter of the decoder: this is common in transfer learning to solve the problem of an insufficient dataset and model over-fitting [36,37,38]. The results show that the transfer learning does enhance the performance with the evaluated error reduction and an improvement of the correlation between model outputs and true results. It should be noted that the dataset in this paper is collected from adults aged 20 to 32 with fifteen males and five females; therefore, the assessment results of the model may be biased in the female population and may give invalid references in the elder population. In this case, the results of the objective evaluation such as psychoacoustic annoyance would be more reliable. Further, the deep learning based algorithm proposed in this paper is a feasible solution for assessing the noise annoyance level; however, the assessment results of characteristic populations are unclear. In order to evaluate characteristic populations, their annoyance data need to be collected and used to train algorithms. Compared with the current work, the next research will concentrate on the performance of deep learning models in assessing the annoyance of populations with different characteristics. In the following research, the authors will follow the next steps to conduct the research. (1) Determining the subjects’ acoustic environment: focusing on the road near the residential area and recording noise information at multiple locations. (2) Quantifying the personal information of subjects: by quantifying subjects’ personal information (such as age, gender, education, and so on), it can clarify the characteristics of different subjects. (3) Subjective experiments, including listening experiments and questionnaire interviews, are used to obtain the evaluation of subjects’ annoyance. (4) Analyzing subjective experimental data: identifying the annoyance level of people with different characteristics. (5) Training assessment models based on subjects’ characteristics information: it should be noted that the input data would include the subject’s personal information and the feature of the noise. By inputting the subjects’ personal information and noise data, the model could predict the annoyance of the characteristic population in a specific environment. In addition, for some extreme cases, the available subjects are small; therefore, in order to evaluate the annoyance induced by road noise accurately and quickly, it still needs the joint efforts of researchers in related fields.

## 6. Conclusions

This paper proposes a deep learning-based objective assessment method for road traffic noise annoyance that can achieve a rapid assessment of road traffic noise annoyance using the amplitude spectrum of traffic noise without incorporating subjects’ personal information or additional acoustic features. To obtain the dataset required for training the model, the authors conduct listening experiments to gather reliable raw data, from which abnormal samples are removed to construct a dataset of the perceived annoyance of road traffic noise.

To evaluate the reliability of the algorithms, artificial neural networks, linear regression based on a principal component analysis, Ridge regression, and Lasso regression are used for comparison. The experimental results show that the evaluation results obtained using deep learning models produce optimal results for the mean absolute error, Pearson’s correlation coefficient, and Spearman’s correlation coefficient. However, in terms of the prediction accuracy in different annoyance intervals, the difference between the maximum and minimum evaluation errors in different intervals for the directly trained model is 1.21, which is less robust. To address this issue, the paper introduces transfer learning to optimize the algorithm through a computer-generated psychoacoustic annoyance degree dataset, preliminary optimization of model parameters to obtain a pre-trained model, and secondary optimization of overall parameters and some parameters on the perceptual noise annoyance degree dataset.

The experimental results show that the model with partial parameter optimization shows a significant improvement in each measure and its accuracy for annoyance intervals with a small sample size has significantly improved. Last but not the least, this study is only conducted for college students, which may lead to certain limitations in the final performance, but this method is still a useful attempt to combine deep learning with noise evaluation. The evaluation results obtained by the deep learning model can be an effective evaluation reference for urban planning, noise management, and relevant noise policy formulation.

## Figures and Tables

**Figure 1 ijerph-20-05199-f001:**
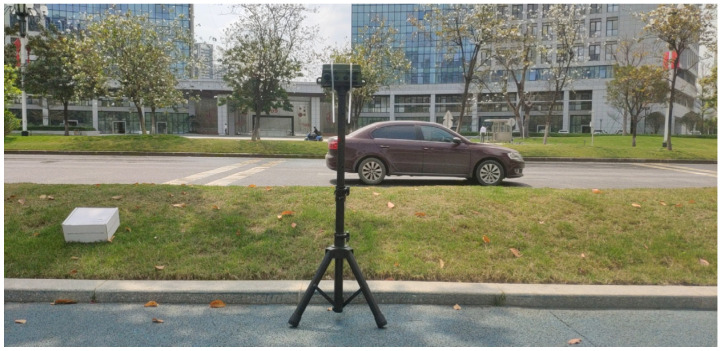
Recording scenes.

**Figure 2 ijerph-20-05199-f002:**
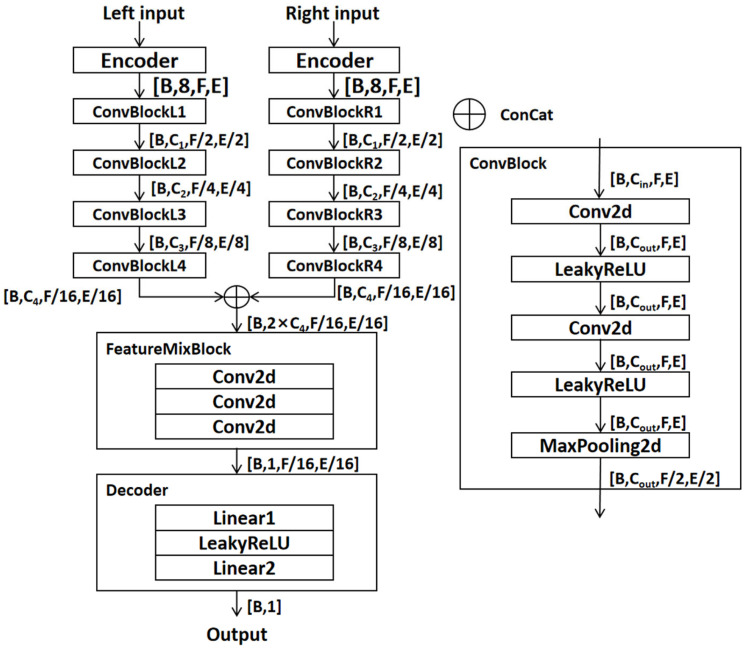
Model architecture. The left is the whole process, and the right is the detail of ConvBlock.

**Figure 3 ijerph-20-05199-f003:**
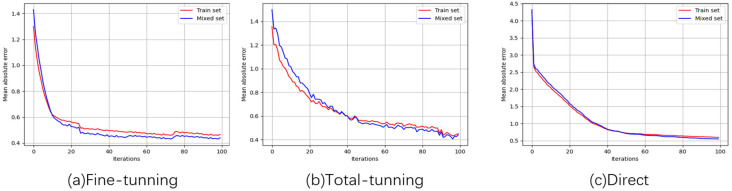
The changes of MAE in deep learning-based model training. (**a**) is the process of fine-tuning; (**b**) shows the variation of MAE in total-tuning training; (**c**) shows the MAE changes of direct.

**Table 1 ijerph-20-05199-t001:** Results after post-processing and annoyance distribution of listening experiment. Adopting the 11-level assessment scale [27], in this paper, the range of annoyance is (0,10).

Annoyance Interval	The Number of Noise Samples
[0,1)	0
[1,2)	1
[2,3)	17
[3,4)	72
[4,5)	95
[5,6)	239
[6,7)	266
[7,8)	186
[8,9)	72
[9,10)	1
Total	949

**Table 2 ijerph-20-05199-t002:** The distribution of psychoacoustic annoyance data. Calculated by the Zwicker method [21]; in this paper, the range of psychoacoustic annoyance is (0,100).

Psychoacoustic Annoyance Interval	The Number of Noise Samples
[0,10]	544
(11,20]	1994
(20,30]	3412
(30,40]	2494
(40,50]	2727
(50,60]	1832
(60,70]	2377
(70,80]	1356
(80,90]	280
(90,100]	9
total	17,025

**Table 3 ijerph-20-05199-t003:** The detailed settings of the model. The left input and the right input will share the same network structure within Encoder, ConvBlock1, ConvBlock2, ConvBlock3, and ConvBlock4, but the weights are not shared.

Network	Layer	Input Size	Output Size	Kernel	Stride	Padding
Encoder	Conv2d	1 × 1499 × 257	8 × 1499 × 257	(1,1)	(1,1)	(0,0)
ConvBlock1	Conv2d + LeakyReLU	8 × 1499 × 257	32 × 1499 × 257	(3,3)	(1,1)	(1,1)
	Conv2d + LeakyReLU	32 × 1499 × 257	32 × 1499 × 257	(3,3)	(1,1)	(1,1)
	Maxpooling2d	32 × 1499 × 257	32 × 749 × 128	(2,2)	(2,2)	(0,0)
ConvBlock2	Conv2d + LeakyReLU	32 × 749 × 128	64 × 749 × 128	(3,3)	(1,1)	(1,1)
	Conv2d + LeakyReLU	64 × 749 × 128	64 × 749 × 128	(3,3)	(1,1)	(1,1)
	Maxpooling2d	64 × 749 × 128	64 × 374 × 64	(2,2)	(2,2)	(0,0)
ConvBlock3	Conv2d + LeakyReLU	64 × 374 × 64	32 × 374 × 64	(3,3)	(1,1)	(1,1)
	Conv2d + LeakyReLU	32 × 374 × 64	32 × 374 × 64	(3,3)	(1,1)	(1,1)
	Maxpooling2d	32 × 374 × 64	32 × 187 × 32	(2,2)	(2,2)	(0,0)
CovBlock4	Conv2d + LeakyReLU	32 × 187 × 32	8 × 187 × 32	(3,3)	(1,1)	(1,1)
	Conv2d + LeakyReLU	8 × 187 × 32	8 × 187 × 32	(3,3)	(1,1)	(1,1)
	Maxpooling2d	8 × 187 × 32	8 × 93 × 16	(2,2)	(2,2)	(0,0)
Concat	Concat	(8 × 93 × 16,8 × 93 × 16)	16 × 93 × 16	None	None	None
FeatureMixBlock	Conv2d	16 × 93 × 16	16 × 93 × 16	(3,3)	(1,1)	(1,1)
	Conv2d	16 × 93 × 16	8 × 93 × 16	(3,3)	(1,1)	(1,1)
	Conv2d	8 × 93 × 16	1 × 93 × 16	(3,3)	(1,1)	(1,1)
Decoder	Linear1 + LeakyReLU	1 × 93 × 16	1 × 93 × 1	16	None	None
	Squeeze	1 × 93 × 1	1 × 93	None	None	None
	Linear2	1 × 93	1 × 1	93	None	None

**Table 4 ijerph-20-05199-t004:** The distribution of mixed set. It is used to evaluate the performance of all algorithms for the assessment of road noise annoyance The data format is the same as Table 1.

Annoyance Interval	The Number of Noise Samples
[2,3)	3
[3,4)	21
[4,5)	28
[5,6)	81
[6,7)	68
[7,8)	53
[8,9)	26
Total	280

**Table 5 ijerph-20-05199-t005:** Comparison of algorithms on different annoyance intervals from different annoyance intervals. Bold indicates the best score in each interval. The range of MAE is [0,10].

Annoyance Intervals	MAE						
	Artificial Neural Network	Linear	Lasso	Ridge	Direct	Total-Tuning	Fine-Tuning
[2,3)	3.22	2.86	3.01	2.89	1.63	0.77	**0.77**
[3,4)	2.61	1.76	1.95	1.83	0.96	0.41	**0.36**
[4,5)	1.66	1.00	1.10	1.01	0.41	0.41	**0.32**
[5,6)	0.79	0.32	0.32	**0.31**	0.42	0.39	0.48
[6,7)	**0.26**	0.55	0.52	0.56	0.57	0.40	0.43
[7,8)	0.76	0.91	0.98	0.99	0.63	0.54	**0.53**
[8,9)	1.71	1.69	1.77	1.82	0.66	0.49	**0.43**
Mean error	0.99	0.82	0.84	0.85	0.57	**0.45**	0.46

**Table 6 ijerph-20-05199-t006:** Evaluation of algorithms’ prediction results with true results from entire dataset. Bold indicates the best score in each case. The range of MAE is [0,10].

Algorithms	Mean Error	PCC	SCC
Artificial Neural Network	0.99	0.46	0.47
Linear Regression	0.82	0.58	0.69
Lasso Regression	0.84	0.54	0.67
Ridge Regression	0.85	0.54	0.67
Direct	0.57	0.87	0.87
Total-tuning	**0.45**	0.92	0.91
Fine-tuning	0.46	**0.93**	**0.92**

## Data Availability

Not applicable.

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
