# Peer review of "Deep Learning-Based Road Traffic Noise Annoyance Assessment"

_ijerph, 2023, doi:10.3390/ijerph20065199_

Round 1

Reviewer 1 Report

Dear

the article deals with the issue of noise analysis using deep learning verified by experiment. The issue is very current with impacts on the health of the population, especially in the immediate vicinity of the transport network.

I have the following comments on the draft text:

• explain the abbreviations MFCC, FFT, PCA and other abbreviations used.

• lines 80 – 92 contain a statement that is not cited in the literature. It is not explained by what methods these conclusions were reached

• there is no reference to what device it was used for, what the error rate was, how the data was divided.

• the experiment was done on the subjective evaluation of mostly students, although the effects are more on older years

• it is not clear how the filters were used and what the FFT was used for.

• Table No. 1 is incomprehensible and not linked to the text

• the distribution of test, training and validation data and their sizes and ratios are not clear.

• clear methods and uses of FFT are not given with the assignment of sample outputs

• there is no description of the deep learning methods that were used, nor is there a reference to the literature.

• the literature should be expanded and supplemented

• it is not clear from tables 1-3 what the data interpret in principle, it is necessary to describe it better

• it is not clear why the stated convolution was chosen and whether the result is optimal for the mean absolute error, including that linear regression was used. Noise data is not linear in nature

• there is a lack of balance, analysis and discussion of another method and the appropriateness of applying it to noise data and to the subjective evaluation of 20 people.

• it is not clear from the text and description how Table 5 was arrived at, what method and approach were used. Nothing is presented in the introduction as to what methods will be used.

• table 6 compares the value and gives some prediction, it is not clear from the text what it is based on, what were the sets for testing, training and validation.

• no discussion is done. The conclusion is not based on the actual text of the article and does not confirm what was already stated in the abstract.

I recommend reject the article and fundamentally reworking, clarifying and supplementing it

best regards

Author Response

Dear reviewer:

       Thank you for your questions and suggestions, we are eager to communicate with you about the problems with this article, here are our responses to the questions.

Issue1: explain the abbreviations MFCC, FFT, PCA and other abbreviations used.

Response: Thanks for your question, we find it hard for readers to understand what it is if the reader does not know these abbreviations before, it can’t be tolerated.

we revised this issue as follows:

  1. In line 59, we revised “MFCC (Mel Frequency Cepstral Coefficient, a kind of speech feature parameter)”.
  2. For time-variant signals, we often use a window to analyze the values within the window, and do FFT (Fast Fourier transform) on these values, this analysis method is called STFT (Short Time Fourier Transform), if we do not use a window to analysis the signal, it is FFT. This is descriptive error and we revise it Immediately.

In line 166, we revised ”STFT (Short Time Fourier Transform, a method to determine the frequency and phase of a sine wave in the local area of a time-varying signal)”.

  1. In line 356, we revised “principal component analysis (PCA, a method to reduce the feature dimensionality while minimizing information loss)”.

Issue 2: lines 80 – 92 contain a statement that is not cited in the literature. It is not explained by what methods these conclusions were reached

Response: We are so grateful for your kind question ,we have reviewed the content between line 80 and line 92 and added more description with the help of your suggestion , here is our revision(Bold for our additions and modifications):

Statement 1: line79-81“Following the transfer learning Strategy[19-20], the method does not require additional input information and keeps the computational complexity of the algorithm unchanged.” Based on transfer learning, we will train the deep learning model on the pre-train dataset and then we train the model on the task dataset. In this process, we don’t change the number of model parameter and the structure of model. Technically, the computational complexity of the algorithm is unchanged.

Statement 2:” Second, compared to the way of increasing the amount of data to obtain features, there is an objective method for calculating the psychoacoustic annoyance of noise[21], and it is only necessary to produce the dataset with the help of a computer, by transferring the psycho-acoustic feature into deep learning model, which does not require large-scale listening experiments.” In this section, we follow the idea of transfer learning, psycho-acoustic annoyance and perceived annoyance have a great correlation and we can calculate the psycho-acoustic annoyance based on Zwicker method, it can be done automatically with computer and do not needsubject listening test.    

Issue 3: there is no reference to what device it was used for, what the error rate was, how the data was divided.

Response: We thank the reviewer for raising this question. We have added the reference to the device we used and here are the revised contents:

  1. 1 line110:” The acquisition equipment song meter4(SM4)[24] was used for road noise recording was song meter4(SM4)”
  2. 2 line123-line115:” the noise was passed through a high-quality sound card (RME Fireface II)[25] for playback and then subjects perceived the noise audio through headphones HD600[26]”
  3. and we revised the description of the error rate in section1 line87-line89:” Based on the experimental results of this paper, the results show that the method using deep learning has reduced the mean absolute error of the prediction by about 30%”.
  4. Finally, the divide method of the pre-training stage was revised in section4.1.1 line327-line329:” the total samples are 17025,12910(75% of the total samples) data are used for training, 1600(10% of the total samples) data for validation, and 2515 (15% of the total samples) data for testing”, the divide method of formal training stage was revised in section4.1.2 line351-line353:” the 949 data from the listening experiment are divided into 669(70% of the total samples), 70(7% of the total samples), and 210(23% of the total samples) for training, validation and testing respectively.”

Issue 4: the experiment was done on the subjective evaluation of mostly students, although the effects are more on older years

Response: Thank you for pointing out this issue and it provides ideas for our next research. From the perspective of algorithms, we can extract road noise features as input and use subjects’ annoyance as a label, it builds a supervised learning dataset suitable for training algorithms, Although the dataset is literally enough for the algorithm, however, there are still some issues need to be mentioned in this paper, we have added the content at the last of section2.2, as follows: “The noise feature data can be used as algorithm input and the subjects’ post-processed annoyance value is used as a label, it builds a supervised learning dataset suitable for training algorithms. Although the data set is literally enough for the algorithm, however, there are still some issues that need to be mentioned: (1) Uneven distribution of subjects, the elder should be considered. Due to the lack of annoyance data about old people, this will lead to changes in subjective annoyance value. (2) The number of subjects. Referring to documents [28-30],20 subjects can basically feedback the annoyance of noise data but the more subjects, the better [17,31-33]. (3) Limitation of laboratory playback. Laboratory playback can recreate the noise well, but it definitely does not recreate the whole of the experience. The laboratory environment does not simulate the subject's situation at the roadside. For wider review and more complete subjective assessment, a qualitative interview is essential [10,17,34-35]. (4) The recording scene is not rich enough. In this paper, the authors just recorded the noise data on the same avenue, noise from other sites should also be taken into consideration”.

Issue5:it is not clear how the filters were used and what the FFT was used for.

Response: Thank you for raising this fatal issue, it is our problem that we didn't describe it clearly. In this paper, we use the amplitude spectrum as algorithm input, the amplitude spectrum contains the amplitude information of the signal at each frequency, to calculate the amplitude spectrum of the noise signal, the FFT is needed. In issue1,we describe the difference between FFT and STFT, We added the description in section2.3 line164-line178: “ In this paper, we select amplitude spectrum as model input. Firstly, an STFT (Short Time Fourier Transform, a method to determine the frequency and phase of a sine wave in the local area of a time-varying signal) was performed on the noise signal to calculate the amplitude value of different frequencies. The process of STFT is as follow:

Where  is represented window function, is noise signal, is the time index,  is the interval with time t, f is the frequency index, the output of STFT is a complex number and then the amplitude spectrum was calculated by the STFT results of noise signal, where the STFT window is hamming window and the window length was 512, the jump length was 256. The calculation of amplitude spectrum is shown as follow:

Where R is the real part of  and Imag means imaginary part of . The function of hamming window is as follow:

Where N is window length minus 1.”  

Issue6: Table No. 1 is incomprehensible and not linked to the text

Response: sorry for our bad description, the table 1 wants to show the overall situation of subjects’ annoyance. In the listening test, we adopt 11-level assessment scale of the ISO 15666:2021 standard and the subjects’ scores are single integer numbers. Each noise will be playback three times, we calculate the mean value to represent the subjects’ annoyance level and this mean value may not be an integer number. Furthermore, we divide the annoyance range [0-10] into different annoyance intervals with an interval size of 1 and let readers know the information about listening test results. For a better understanding of the table 1,we have revised the following content at section 2.3 line180-line185” Due to calculating the mean value of annoyance, the annoyance value is no longer an integer type ,so we divided the annoyance range [0-10] into different annoyance intervals with an interval size of 1.949 samples were obtained and the distribution of the samples is shown in Table 1. The level of annoyance caused by traffic noise is usually high, few samples fall in [0,3) interval and the most noise sample fall in [3,9).” 

Issue7: the distribution of test, training and validation data and their sizes and ratios are not clear.

Response: We thank the reviewer for raising this question. In order to highlight the distribution of test, training and validation data we have updated the description and adjusted the structure of the article as follow : (1)the divide method of pre-training stage was revised in section4.1.1 line327-line329:” the total samples are 17025,12910(75% of the total samples) data are used for training, 1600(10% of the total samples) data for validation, and 2515 (15% of the total samples) data for testing”

(2) The divide method of formal training stage was revised in section4.1.2 line351-line353:” the 949 data from the listening experiment are divided into 669(70% of the total samples), 70(7% of the total samples), and 210(23% of the total samples) for training, validation and testing respectively.”

Issue8: clear methods and uses of FFT are not given with the assignment of sample outputs

Response: We are so grateful for your kind question. In the section2.3 (lines 164-178), we added the methods and uses of FFT, as follows: “In this paper, we select amplitude spectrum as model input. Firstly, an STFT (Short Time Fourier Transform, a method to determine the frequency and phase of a sine wave in the local area of a time-varying signal) was performed on the noise signal to calculate the amplitude value of different frequencies. The process of STFT is as follow:

Where  is represented window function, is noise signal, is the time index,  is the interval with time t, f is the frequency index, the output of STFT is a complex number and then the amplitude spectrum was calculated by the STFT results of noise signal, where the STFT window is hamming window and the window length was 512, the jump length was 256. The calculation of amplitude spectrum is shown as follow:

Where R is the real part of  and Imag means imaginary part of . The function of hamming window is as follow:

Where N is window length minus 1.”

Issue9:there is no description of the deep learning methods that were used, nor is there a reference to the literature.

Response: We thank the reviewer for raising this question, we have made the following changes to the section3.1 line227-line252, as follows:” With the development of deep learning, a large number of researchers try to use deep learning on its own tasks and many researchers have successfully applied deep learning algorithms in lots of fields such as image segmentation [41], speech enhancement [42], hearing aids [43], traffic prediction [44] and so on. In this process, many classic deep learning model architectures have been gradually created, among which the most widely used and effective model is UNet [45], almost all of tasks achieved good results by using UNet architecture or modifying the UNet architec-ture according to the requirements of the tasks [41,44,45]. The model extracts deep features of the data by adding a convolutional downsampling module[41] to the basic convolutional neural network[46] and recovers the dimensionality of the data through a convolutional upsampling[41] module.

In this paper, we use Unet as the basic model and the reasons are as follows:(1) Unet and its variants version have been used in many research areas with good results, it can provide ideas for our research[41,44,45,47]. (2) UNet is a kind of a deep convolutional neural network [46], the convolutional operation[46] is commonly used in deep learning model, by using convolutional operation, maxpooling operation and activate function, the network can implement nonlinear transformations to get the feature map.(3) Unet is a mature deep learning network architecture. Compared with other model structures, Unet's structure is relatively simple, it has more online open source information and easy to reproduce [48]. In this paper, by modifying the structure according to task requirement, we enable end-to-end evaluation of traffic noise based on deep learning model. The model does not need to recover the output dimension of the data but only needs to use the convolutional downsampling module for feature extraction of road noise to get feature map. The Feature fusion module will fuse the extracted features and then use fully connected layers network [46] as decoder to output the final result. The model architecture is shown in Figure 2.”

Issue10: the literature should be expanded and supplemented

Response: Thank you for your valuable comments, we expanded the literature and its description and made the following changes:

  1. section 2.2 line137-151:” The noise feature data can be used as algorithm input and the subjects’ post-processed annoyance value is used as label, it builds a supervised learning dataset suitable for training algorithms. Although the data set is literally enough for algorithm, however, there are still some issues need to be mentioned: (1) Uneven distribution of subjects, the elder should be considered. Due to the lack of annoyance data about old people, this will lead to changes in subjective annoyance value. (2) The number of subjects. Referring to documents [28-30],20 subjects can basically feedback the annoyance of noise data but the more subjects, the better [17,31-33]. (3) Limitation of laboratory playback. Laboratory playback can recreate the noise well, but it definitely does not recreate the whole of the experience. The laboratory environment does not simulate the subject's situation at the roadside. For wider review and more complete subjective assessment, a qualitative interview is essential [10,17,34-35]. (4) The recording scene is not rich enough. In this paper, the authors just recorded the noise data on the same avenue, noise from other sites should also be taken into consider.”
  2. 1 line227-252:” With the development of deep learning, large number of researchers try to use deep learning on its own tasks and many researchers have successfully applied deep learning algorithms in lots of fields such as image segmentation [41], speech enhancement [42], hearing aids [43], traffic prediction [44] and so on. In this process, many classic deep learning model architectures have been gradually created, among which the most widely used and effective model is UNet [45], almost all of tasks achieved good results by using UNet architecture or modifying the UNet architec-ture according to the requirements of the tasks [41,44,45]. The model extracts deep features of the data by adding a convolutional downsampling module[41] to the basic convolutional neural network[46] and recovers the dimensionality of the data through a convolutional upsampling[41] module. In this paper, we use Unet as the basic model and the reasons are as follows:(1) Unet and its variants version have been used in many research areas with good re-sults, it can provide ideas for our research[41,44,45,47]. (2) UNet is a kind of a deep convolutional neural network [46], the convolutional operation[46] is commonly used in deep learning model, by using convolutional operation, maxpooling opera-tion and activate function, the network can implement nonlinear transformations to get the feature map.(3) Unet is a mature deep learning network architecture. Com-pared with other model structures, Unet's structure is relatively simple, it has more online open source information and easy to reproduce [48]. In this paper, by modi-fying the structure according to task requirement, we enable end-to-end evaluation of traffic noise based on deep learning model. The model does not need to recover the output dimension of the data but only needs to use the convolutional downsampling module for feature extraction of road noise to get feature map. The Feature fusion module will fuse the extracted features and then use fully connected layers network[46] as decoder to output the final result. The model architecture is shown in Figure 2.”

Issue11:it is not clear from tables 1-3 what the data interpret in principle, it is necessary to describe it better

Response: We are so grateful for your kind question. We have revised the descriptions about table1-3 and the changes are as follow:

  1. 3 line180-line185:” Due to calculating the mean value of annoyance, the annoyance value is no longer an integer type, so this paper divided the annoyance range [0-10] into different annoyance intervals with an interval size of 1.949 samples were obtained and the distribution of the samples is shown in Table 1. The level of annoyance caused by traffic noise is usually high, few samples fall in [0,3) interval and the most noise sample fall in [3,9).”
  2. Section 2.4 line202-line207:” The distribution of the data samples is shown in Table 2, the samples should be evenly distributed in each interval as far as possible, where the Psycho-acoustic annoyance interval means the Psycho-acoustic annoyance value in different interval and the numbers means the amount of samples in the corresponding interval. The Psycho-acoustic is continuous value and the most data fall in (0,90]”.
  3. 2.1 line341-line344:” To avoid analysis errors due to the small amount of test data, this paper chooses the best model in validation and analysis the performance on the mixed set (mix test data with validation data). The distribution of the mixed dataset is shown in Table 3. In this mixed set, no value falls in [0,2) and [9,10).”

Issue12: it is not clear why the stated convolution was chosen and whether the result is optimal for the mean absolute error, including that linear regression was used. Noise data is not linear in nature

Response: We thank the reviewer for raising this question, we have revised the description and added the content to explain: (1)why the stated convolution was chosen, (2)whether the result is optimal for the mean absolute error and (3)why linear regression was used.

  1. In section 3.1 line238-252, to illustrate the reasons for using stated convolution operations,we added “In this paper, we use Unet as the basic model and the reasons are as follows:(1) Unet and its variants version have been used in many research areas with good re-sults, it can provide ideas for our research[41,44,45,47]. (2) UNet is a kind of a deep convolutional neural network [46], the convolutional operation[46] is commonly used in deep learning model, by using convolutional operation, maxpooling opera-tion and activate function, the network can implement nonlinear transformations to get the feature map.(3) Unet is a mature deep learning network architecture. Com-pared with other model structures, Unet's structure is relatively simple, it has more online open source information and easy to reproduce [48]. In this paper, by modi-fying the structure according to task requirement, we enable end-to-end evaluation of traffic noise based on deep learning model. The model does not need to recover the output dimension of the data but only needs to use the convolutional downsampling module for feature extraction of road noise to get feature map. The Feature fusion module will fuse the extracted features and then use fully connected layers network[46] as decoder to output the final result. The model architecture is shown in Figure 2.”
  2. In section4.1.2 line325-line329, we explained why MAE was chosen and added “we select mean absolute error(MAE) as an evaluation standard , the MAE can visu-ally feedback the average error between the predicted result and the real result, the evaluation weight for each error is equal, it is less affected by anomaly samples[50], it suits for evaluating the overall performance of the model.”
  3. In section4.2.2 line360-line366, we gave the reasons for using linear regression and added” Artificial neural network has nonlinear fitting capabilities,in this paper, the built ar-tificial neural network consists of five linear fully connected layers and the output of each layer is activated by LeakyReLU. The regression algorithms are commonly analysis methods[16,17,50] and It is difficult to directly figure out the relationship of post-processing features ,so this paper uses linear regression, Lasso regression and Ridge regression as the basic comparison algorithm to evaluate the noise annoyance.”

Issue13:there is a lack of balance, analysis and discussion of another method and the appropriateness of applying it to noise data and to the subjective evaluation of 20 people.

Response: We are so grateful for your kind question. We have revised the content about the discussion of different method, analysis of subjective evaluation and the cause why this article try to train different method on this dataset, the added descriptions are as follow:

  1. analysis and discussion of transfer learning method: section1 line68-71,“However, due to the long data acquisition period of subjective listening experiments, the available sample size is small at this stage, and problems such as overfit-ting easily occur in the training process [18]. To avoid this problem, this paper introduces transfer learning to solve this problem.”line81-87“Following the transfer learning Strategy [19-20], the method does not require additional input information and keeps the computational complexity of the algorithm unchanged. Second, compared to the way of increasing the amount of data to obtain features, there is an objective method for calculating the psychoacoustic annoyance of noise[21], and it is only necessary to produce the dataset with the help of a computer, by using of transferring the psycho-acoustic feature into deep learning model, which does not require large-scale listening experiments.”

section5 line444-line453 “This paper adopts three strategies to train the deep learning network, one of which is directly trained on the listening experiment dataset with better performance than commonly used machine learning algorithms and the other two, called total-tunning and fine-tunning respectively, are pre-trained on the psychoacoustic dataset first and then trained on listening experiment dataset, where total-tunning will optimize all parameters and fine-tunning just optimizes the parameter of de-coder, this is common in transfer learning to solve the problem of insufficient dataset and model overfitting[36-38]. The results show that the transfer learning does enhance the performance with evaluated error reduction and improvement of correlation between model outputs and true results.”

  1. Analysis and Discussion of why this article try to train different method on this dataset:Section section2.2 line137-line139:” The noise feature data can be used as algorithm input and the subjects’ post-processed annoyance value is used as label, it builds a supervised learning dataset suitable for training algorithms”

section5 line434-line438” From the perspective of algorithm training, the amplitude spectrums of noises are input data and subjects’ annoyance scores are used as label, it built a supervised learning dataset suitable for training algorithms and the results of the different algorithms show the feasibility of training on this dataset.”

  1. Analysis and discussion of the subjective evaluation:section2.2 line139-line151” Although the data set is literally enough for algorithm, however, there are still some issues need to be mentioned: (1) Uneven distribution of subjects, the elder should be considered. Due to the lack of annoyance data about old people, this will lead to changes in subjective annoyance value. (2) The number of subjects. Referring to documents [28-30],20 subjects can basically feedback the annoyance of noise data but the more subjects, the better [17,31-33]. (3) Limitation of laboratory playback. Laboratory playback can recreate the noise well, but it definitely does not recreate the whole of the experience. The laboratory environment does not simulate the subject's situation at the roadside. For wider review and more complete subjective assessment, a qualitative interview is essential [10,17,34-35]. (4) The recording scene is not rich enough. In this paper, the authors just recorded the noise data on the same avenue, noise from other sites should also be taken into consider.”

Section5 line433-line444” To train the model, 20 subjects (15 males and 5 females, aged between 20 and 32 years) were paid for joining in a listening experiment. From the perspective of algorithm training, the amplitude spectrums of noises are input data and subjects’ annoyance scores are used as label, it built a supervised learning dataset suitable for training algorithms and the results of the different algorithms show the feasibility of training on this dataset. But it should be noted that the 20 subjects can basically feedback the annoyance of noise data [28-30] but the more subjects join in the listening experiment, the results will be closer to the real world[31-33]. The distribution of subjects is equally important, lacking the old’s subjective results, this approach may fail in the elderly population. In addition, it is difficult to reproduce the complete experience with recreate noise in a laboratory environment, for more ac-curate evaluation, a qualitative interview is needed.”

Issue14:it is not clear from the text and description how Table 5 was arrived at, what method and approach were used. Nothing is presented in the introduction as to what methods will be used.

Response: We thank the reviewer for raising this question and sorry for our writing error. The table number was written incorrectly, the table 5 is actually the table 6 in this paper, we revised it and checked formulas, tables and figures throughout the article to avoid repeating this error.

Issue15: table 6 compares the value and gives some prediction, it is not clear from the text what it is based on, what were the sets for testing, training and validation.

Response: We are so grateful for your kind question. We adjusted the article structure and added section 4.2.2 to elaborate the compares come from, the revised content is as follow :

  1. Section 4.2.2 line351-line353 :” To compare the performance of the algorithms, artificial neural networks[17], linear regression [17], Lasso regression, and Ridge regression [16], and directly trained models, denoted as direct, are used as comparison algorithms.”
  2. Section 4.2.2 line377-line381:” The MAE are chosen to measure the error between evaluation results and true results, Pearson's correlation coefficient (formula 10, denoted as PCC)[50], and Spearman's correlation coefficient (formula 11, denoted as SCC)[50] are chosen to measure the correlation between the evaluation results of the model and the true results.”
  3. And we added section 4.2.1 line348-line356 to describe the dataset, the added content is as follow:”

4.2. formal training stage

4.2.1 formal training dataset setup

At the formal training stage, based on the pre-trained model, the model parameters will be optimized on listening experiment dataset (section2.3,table1), the 949 data from the listening experiment are divided into 669(70% of the total samples), 70(7% of the total samples), and 210(23% of the total samples) for training, validation and testing respectively. To avoid analysis errors due to the small amount of test data, this paper chooses the best model in validation and analysis the performance on the mixed set (mix test data with validation data). The distribution of the mixed data set is shown in Table 3. In this mixed set, no value falls in [0,2) and [9,10).”

Issue16:no discussion is done. The conclusion is not based on the actual text of the article and does not confirm what was already stated in the abstract.

Response: We thank the reviewer for raising this issue. we have added the discussion as section5 and revised the analysis in section4.2.2 line384-line386,line414-line416. The changes are as

Follow:”

5.Discussion

In this paper, a deep learning-based objective assessment method for road traffic noise annoyance is proposed. Different from [16] and [17], this method does not need subjects' personal information or additional acoustic features, by inputting the amplitude spectrum, this assessment method can rapidly evaluate the annoyance level of road noise. To train the model, 20 subjects (15 males and 5 females, aged between 20 and 32 years) were paid for joining in a listening experiment. From the perspective of algorithm training, the amplitude spectrums of noises are input data and subjects’ annoyance scores are used as label, it built a supervised learning dataset suitable for training algorithms. But it should be noted that the 20 subjects can basically feedback the annoyance of noise data [28-30] but the more subjects join in the listening experiment, the results will be closer to the real world[31-33]. The distribution of subjects is equally important, lacking the old’s subjective results, this approach may fail in the elderly population. In addition, it is difficult to reproduce the complete experience with recreate noise in a laboratory environment, for more accurate evaluation, a qualitative interview is needed.

This paper adopts three strategies to train the deep learning network, one of which is directly trained on the listening experiment dataset with better performance than commonly used machine learning algorithms and the other two, called total-tunning and fine-tunning respectively, are pre-trained on the psychoacoustic dataset first and then trained on listening experiment dataset, where total-tunning will optimize all parameters and fine-tunning just optimizes the parameter of de-coder, this is common in transfer learning to solve the problem of insufficient dataset and model overfitting[36-38]. The results show that the transfer learning does enhance the performance with evaluated error reduction and improvement of correlation between model outputs and true results.

Further, the deep learning based algorithm proposed in this paper is a feasible solution for assessing the noise annoyance level and in order to adequately assess the annoyance level induced by road noise, it needs the joint efforts of researchers in related fields.”

Section4.2.2 line384-line386:” In terms of the entire mixed dataset, comparing to the mean absolute error of regression algorithms and neural network, the mean error of Direct is just 0.57, the Direct reduces the mean error by about 30%”

Section4.2.2 line414-line419:” total-tunning has the smallest evaluation error but it is close to Fine-tunning's. On the other hand, comparing to the MAE of Direct ,the Fine-tunning is 0.45, Fi-ne-tunning reduces this error by about 30% and the evaluation results obtained by optimizing with the Fine-tunning strategy have the largest correlation with the true results, the Pearson correlation coefficient and the Spearman correlation coefficient reach 0.93 and 0.92, comparing to Direct, it improves about 6% and 5% respectively”

Thanks again for your kind questions and comments. We are looking forward to hearing from you!

Reviewer 2 Report

Traffic noise is an important problem in many parts of the world. The analysis of the data seems quite adequate. The paper includes a good description of the methods used.

I feel, however, that there should be something more on further research. The whole work is based on field measurements on one site. The subjective assessment is based on laboratory assessment by 20 subjects between 20 and 32 years of age rating the noise on a single numbers scale.

The authors deserve praise for also using psychoacoustic parameters. And it seems that traditional acoustical spectra have also been measured, althoughI don't see it presented.

The paper can stand on its own, as the authors have been clear on what they have done. I would like, however, a somewhat wider view, using different apporaches on subjective evaluations. Laboratory playback can recreate the sound very well, but it definitely does not recreate the whole of the experience. Noise from other sites may well give different results. And a wider group of listeners may give another result. These points should be mentioned. And it would seem that a wider range of data would be well suited to an AI approach. It is possible, even likely, that a similar study in another culture could give different results.

One possible reference for a wider approach both regarding the physical and the perceived characteristics of noise are given below. Even though some of the work is for transport noise other than road traffic noise, the apporach may be useful. Qualitative interviews might give a more complete subjective assessment.

Tore Fodnes Killengreen & Sigmund Olafsen Response to noise and vibration from roads and trams in Norway, ICA 2022, Gyejongyu, Korea

Author Response

Dear reviewer:

       Thank you for your comments on this article. We further realize the shortcomings of our work with your suggestions, which provide us with new thoughts for the next research. We have added the content to illustrate the deficiencies of this paper and the changes are as follows:”

Issue1:”I would like, however, a somewhat wider view, using different approaches on subjective evaluations. Laboratory playback can recreate the sound very well, but it definitely does not recreate the whole of the experience. Noise from other sites may well give different results. And a wider group of listeners may give another result. These points should be mentioned. And it would seem that a wider range of data would be well suited to an AI approach. It is possible, even likely, that a similar study in another culture could give different results.”

For this issue, we summarized the main deficiencies of the current work as follows:

  • Limitation of laboratory playback.
  • The recording scene is not rich enough.
  • Uneven distribution of subjects
  • The number of subjects.

Issue2:” One possible reference for a wider approach both regarding the physical and the perceived characteristics of noise are given below. Even though some of the work is for transport noise other than road traffic noise, the approach may be useful. Qualitative interviews might give a more complete subjective assessment”

For this issue, we knew the importance of detailed subjective evaluation method and supplemented relevant literature.

Combining issue 1 and issue 2, we revised the corresponding content and the changes are as follow:

  1. Section 2.2 line136-line152:” A total of 20 subjects, 15 males and 5 females, aged between 20 and 32 years, were invited to participate in the listening experiment. All subjects had normal hearing. The subjects were paid for completing the experiment. The noise feature data can be used as algorithm input and the subjects’ post-processed annoyance value is used as a label, it builds a supervised learning dataset suitable for training algorithms. Although the data set is literally enough for the algorithm, however, there are still some issues need to be mentioned: (1) Uneven distribution of subjects, the elder should be considered. Due to the lack of annoyance data about old people, this will lead to changes in subjective annoyance value. (2) The number of subjects. Referring to documents [28-30],20 subjects can basically feedback on the annoyance of noise data but the more subjects, the better [17,31-33]. (3) Limitation of laboratory playback. Laboratory playback can recreate the noise well, but it definitely does not recreate the whole of experience. The laboratory environment does not simulate the subject's situation at the roadside. For a wider review and more complete subjective assessment, a qualitative interview is essential [10,17,34-35]. (4) The recording scene is not rich enough. In this paper, the authors just recorded the noise data on the same avenue, noise from other sites should also be taken into consideration.”
  2. Section 5 line515-line516:” To train the model, 20 subjects (15 males and 5 females, aged between 20 and 32 years) were paid for joining in a listening experiment. From the perspective of algorithm training, the amplitude spectrums of noises are input data and subjects’ annoyance scores are used as a label, it built a supervised learning dataset suitable for training algorithms and the results of the different algorithms show the feasibility of training on this dataset. But it should be noted that the 20 subjects can basically feedback on the annoyance of noise data [28-30] but the more subjects join in the listening experiment, the results will be closer to the real world[31-33]. The distribution of subjects is equally important, lacking the old’s subjective results, this approach may fail in the elderly population. In addition, it is difficult to reproduce the complete experience with recreate noise in a laboratory environment, for more ac-curate evaluation, a qualitative interview is needed.”

In addition, there are some details that need to be clarified to you:

  1. We modified the article’s structure and added section 4.1.1, section 4.1.2 to introduce the dataset for algorithms training.
  2. We add a discussion (section 5) to introduce the work and point out the shortcomings of the work.
  3. We did not do a specific spectral analysis and only used the amplitude spectrum of the signal as different algorithms’ input.

Thanks again for your comments and we are looking forward to hearing from you!

Round 2

Reviewer 1 Report

Dear

The article has been supplemented and partially improved. For clarification, it would be appropriate to add the following points

- Add recommendations and clarifications if only student data were used and what would be the prerequisites for changes in older people or defined in an age interval. The same applies to women, who perceive sound a little better than men. There is no mention of it

- Better explain laboratory and field measurements, as they can differ significantly due to their measurement parameters and subsequent results

- UNet was used for deep learning, which is referenced in the form of literature. A precise description of the used convolutional neural network would be appropriate, such as - how many layers, how many neurons, what was the error rate, including graphs of some learning and training outputs etc.

- It was specified that the number of samples for training and testing was in the 75/25 format. It would be useful to state what it would look like for a proposed UNet for 60/40 and 90/10 ratios. In the discussion, it can be evaluated how it would turn out if the data were not only from students, but also from older years, including women. The conclusions for psychoacoustics would thus be significantly stronger and more accurate

- A section on the discussion was added, in which the next procedure, improvement, addition or other potential measurements, etc., should be indicated, and it was clear that if there are few samples, further detailed research needs to be done and the conclusions improved etc.

- From tables 1, 2, 3 it should be immediately obvious what was done. The name "numbers" is insufficient, the question is "numbers of what". Even a separate table must have some telling value.

I recommend that the article be clarified and supplemented

best regards

Author Response

Dear reviewer:

       Thank you for your questions and suggestions. This article has made great progress with your help! Following your suggestion, we revised the article in detail. we are eager to communicate with you about the problems in this article, here are our responses to the issues.

Issue1:”Add recommendations and clarifications if only student data were used and what would be the prerequisites for changes in older people or defined in an age interval. The same applies to women, who perceive sound a little better than men. There is no mention of it”

Response: We sincerely appreciate your thoughtful suggestions.Although limited by objective conditions (human and material resources), this research experiment is only aimed at college students, which may lead to limitations in the final performance. We also thank the reviewer for pointing out this issue and we have added relevant explanations in the abstract(line29-30), discussion(section 5 line471-line484) and conclusion(section 6 line539-line542). However, we think this method is still an useful attempt to combine deep learning with noise assessment. The changes are as follows:

  • abstract(line29-line30):” Although the model trained on college students' data has some limitations, it is still an useful attempt to apply deep learning to noise assessment.”
  • Disucssion(section 5 line471-line484):” The distribution of subjects is equally important, lacking the old’s subjective results, this approach may fail in the elderly population. For the elderly who did not join in the listening experiment and women with a small proportion of subjects, the fol-lowing matters need to take into consideration: (1) Determine the evaluation subjects according to the specific scenario. Students and young people are the main subjects for the traffic roads near the school. In residential areas, subjects of all ages should take part in the subjective assessment, analysis is carried out according to age [17]. In aging communities, older groups need to be prioritized as subjects for assessment. (2) Based on subjects’ characteristics to choose subjects. Comparing to young people, the old people often suffer from hearing loss and the sensitivity to noise will be re-duced[54], they would give a lower annoyance level than young people. Considering the hearing differences between men and women[55], men tend to lose their ability to hear at a higher frequency level and women who lose their hearing in the lower-level frequencies. The noise with energy concentrated on high frequency may make women feel more upset and the low-frequency noise will make men feel uncom-fortable.”
  • Conclusion(section 6 line539-line542):” Last but not the least, this study is only conducted for college students, which may lead to certain limitations in the final performance, but this method is still an useful attempt to combine deep learning with noise evaluation. The evaluation results obtained by the deep learning model can be an effective evaluation reference for urban planning, noise management, and relevant noise policy formulation.”

Issue2:”Better explain laboratory and field measurements, as they can differ significantly due to their measurement parameters and subsequent results”

Response: Thank you for your suggestion and we have added more details about listening  experiments and noise recording. The changes are as follows:

  • 1 line107-line114:” the road traffic flow was concentrated in the range of 400 to 800 vehicles/hour, the wind speed was less than 1m/s, the temperature was 17℃-25℃. In order to reduce the impact of reflected sound on recording, ensure that the recording equipment does not contain large reflectors (such as walls) within 10 meters. If the ambient sound contains too many additional sound sources, such as the sound of birds and the sound of pedestrians, the recording needs to be stopped in time and the recording results should not be used in the listening experiment.”
  • 2 line125-line127,line135-line138:”The experiments were conducted in an audiometric room(6.78×3.51×2.26 (m)) with a background noise of less than 25dBA and the walls and floors of the audiometric room are covered with sound-absorbing materials ”” each listening session was strictly limited to 40 minutes and a total of 14 rounds are conducted. To ensure an accurate assessment of noise, subjects are allowed to stop and rest for 2-3 minutes during the listening experiments. The same subject conducts listening experiment at the same time of the day.”

Issue3:”UNet was used for deep learning, which is referenced in the form of literature. A precise description of the used convolutional neural network would be appropriate, such as - how many layers, how many neurons, what was the error rate, including graphs of some learning and training outputs etc.”

Response: Thanks for your kind suggestion and we added table3(section3.2 line316) to describe the details of the Unet model proposed in this paper:

“Table 3. The detailed settings of model. The left input and the right input will share the same network structure within Encoder, ConvBlock1, ConvBlock2, ConvBlock3, ConvBlock4 but the weights are not shared.

Network

layer

input size

output size

kernel

stride

padding

Encoder

Conv2d

1×1499×257

8×1499×257

(1,1)

(1,1)

(0,0)

ConvBlock1

Conv2d+LeakyReLU

8×1499×257

32×1499×257

(3,3)

(1,1)

(1,1)

Conv2d+LeakyReLU

32×1499×257

32×1499×257

(3,3)

(1,1)

(1,1)

Maxpooling2d

32×1499×257

32×749×128

(2,2)

(2,2)

(0,0)

ConvBlock2

Conv2d+LeakyReLU

32×749×128

64×749×128

(3,3)

(1,1)

(1,1)

Conv2d+LeakyReLU

64×749×128

64×749×128

(3,3)

(1,1)

(1,1)

Maxpooling2d

64×749×128

64×374×64

(2,2)

(2,2)

(0,0)

ConvBlock3

Conv2d+LeakyReLU

64×374×64

32×374×64

(3,3)

(1,1)

(1,1)

Conv2d+LeakyReLU

32×374×64

32×374×64

(3,3)

(1,1)

(1,1)

Maxpooling2d

32×374×64

32×187×32

(2,2)

(2,2)

(0,0)

CovBlock4

Conv2d+LeakyReLU

32×187×32

8×187×32

(3,3)

(1,1)

(1,1)

Conv2d+LeakyReLU

8×187×32

8×187×32

(3,3)

(1,1)

(1,1)

Maxpooling2d

8×187×32

8×93×16

(2,2)

(2,2)

(0,0)

Concat

Concat

(8×93×16,8×93×16)

16×93×16

None

None

None

FeatureMixBlock

Conv2d

16×93×16

16×93×16

(3,3)

(1,1)

(1,1)

Conv2d

16×93×16

8×93×16

(3,3)

(1,1)

(1,1)

Conv2d

8×93×16

1×93×16

(3,3)

(1,1)

(1,1)

Decoder

Linear1+LeakyReLU

1×93×16

1×93×1

16

None

None

Squeeze

1×93×1

1×93

None

None

None

Linear2

1×93

1×1

93

None

None

“.

And we add the figure3 to show the variation of MAE during deep learning based model training(section4.2.2 line401-line410):” Figure3 shows the MAE changes of deep learning based methods during algo-rithms training. The initial error of the direct method is greater than Fine-tunning and Total-tunning, this may be affected by transfer learning, Fine-tunning and To-tal-tunning perceived the characteristics of psychoacoustic annoyance in advance, the error of Fine-tunning and Total-tunning in each iteration is always less than Direct.

Figure 3. The changes of MAE in deep learning based model training. (a) is the process of Fine-tunning;(b) shows the variation of MAE in total-tunning training; (c)shows the MAE changes of Direct

Issue4:”It was specified that the number of samples for training and testing was in the 75/25 format. It would be useful to state what it would look like for a proposed UNet for 60/40 and 90/10 ratios. In the discussion, it can be evaluated how it would turn out if the data were not only from students, but also from older years, including women. The conclusions for psychoacoustics would thus be significantly stronger and more accurate”

Response: Thanks for your kind question and we revised the section4.1.1 to explain why training and testing was in the 75/25 format:” Generally, for machine learning and deep learning datasets, the number of training sets is 70-80% of the total samples and the number of samples in the validation and test sets is 20-30% of the total samples [50-51]. Take the MNIST [52] for example, which is a common dataset in machine learning as well as deep learning, this dataset has a training set of 55,000(78% of total samples), a validation set of 5,000(8% of total samples) and a test set of 10,000(14% of total samples), if the samples in the training set are not sufficient, it will lead to overfitting phenomenon[53] and if the samples in the training set are overdose, It is difficult to measure the ability of the model to handle unknown samples in the future.”and the revised contents about discussion are as follows:” It should be noted that the dataset in this paper is collected from adults aged 20 to 32 with 15 males and 5 females, the assessment results of the model may be biased in the female population and may give invalid references in the elder population. In this case, the results of objective evaluation of psychoacoustic annoyance would be more reliable.”.

Issue5:”A section on the discussion was added, in which the next procedure, improvement, addition or other potential measurements, etc., should be indicated, and it was clear that if there are few samples, further detailed research needs to be done and the conclusions improved etc.”

Response: Thank you for your supplement to the article. We added the content of details about futher researches and the changes are in the last paragraph of the discussion:” Further, the deep learning based algorithm proposed in this paper is a feasible solution for assessing the noise annoyance level, but there are still inadequacies. In the following research, the authors will consider more abundant road noise scenes such as noise scene near urban railway site and roads with speed limitation. In addition, the subjects for listening experiments are determined by the type of person on the research road and more participants are required. Quantitative interview is needed in the next subjective evaluation experiment, by quantifying the personal information of the subjects (such as age, sex, education level, etc.) and the feedback information of the subjects under the specific road noise environment, the deep learning model can take into account the characteristics of the subjects while capture the noise feature, so that the deep learning model can predict the corresponding evaluation results according to the noise scene and the characteristics of the subjects.  and In order to evaluate the annoyance induced by road noise accurately and quicklyin order to adequately assess the annoyance level induced by road noise, it still needs the joint efforts of researchers in related fields.”

Issue6:”From tables 1, 2, 3 it should be immediately obvious what was done. The name "numbers" is insufficient, the question is "numbers of what". Even a separate table must have some telling value.”

Response: Thanks for your suggestion and we found it is hard for readers to understand the meaning of “numbers” immediately. We used “The number of noise samples” instead of “numbers” to let the readers know the number of noise samples in different annoyance intervals.

Thanks again for your kind questions and comments. We are looking forward to hearing from you!